# Measurement of Microcystin Activity in Human Plasma Using Immunocapture and Protein Phosphatase Inhibition Assay

**DOI:** 10.3390/toxins14110813

**Published:** 2022-11-21

**Authors:** Brady R. Cunningham, Rebekah E. Wharton, Christine Lee, Mike A. Mojica, Logan C. Krajewski, Shirley C. Gordon, Adam M. Schaefer, Rudolph C. Johnson, Elizabeth I. Hamelin

**Affiliations:** 1Division of Laboratory Sciences, National Center for Environmental Health, Centers for Disease Control and Prevention, Atlanta, GA 30341, USA; 2Division of Foodborne, Waterborne, and Environmental Diseases, National Center Emerging and Zoonotic Infectious Diseases, Centers for Disease Control and Prevention, Atlanta, GA 30333, USA; 3Christine E. Lynn College of Nursing, Florida Atlantic University, Boca Raton, FL 33431, USA; 4Abt Associates Inc., Fort Myers, FL 33916, USA

**Keywords:** cyanobacteria, harmful algal bloom, microcystin, immunocapture, Florida, conjugate, automation, cysteine, glutathione

## Abstract

Microcystins are toxic chemicals generated by certain freshwater cyanobacteria. These chemicals can accumulate to dangerous levels during harmful algal blooms. When exposed to microcystins, humans are at risk of hepatic injury, including liver failure. Here, we describe a method to detect microcystins in human plasma by using immunocapture followed by a protein phosphatase inhibition assay. At least 279 microcystins have been identified, and most of these compounds share a common amino acid, the Adda side chain. We targeted this Adda side chain using a commercial antibody and extracted microcystins from human samples for screening and analysis. To quantitate the extracted microcystins, we fortified plasma with microcystin-LR, one of the most well-studied, commonly detected, and toxic microcystin congeners. The quantitation range for the detection of microcystin in human plasma using this method is 0.030–0.50 ng/mL microcystin-LR equivalents. This method detects unconjugated and conjugated forms (cysteine and glutathione) of microcystins. Quality control sample accuracies varied between 98.9% and 114%, with a precision of 7.18–15.8%. Finally, we evaluated plasma samples from a community health surveillance project of Florida residents living or working near harmful algae blooms.

## 1. Introduction

Harmful algal blooms commonly occur in eutrophic bodies of water. Under these conditions, certain freshwater cyanobacteria, such as *Microcystis aeruginosa*, can form and release cyclic peptide hepatotoxins, called microcystins (MCs), into the water [1,2,3,4]. Humans exposed to MCs might experience a range of symptoms, from gastrointestinal distress, including abdominal pain, diarrhea, nausea, and vomiting, to more severe conditions, including pneumonia or liver failure [3,5]. In addition to these symptoms, MCs are also potent tumor promoters [6,7,8,9]. Because of these effects, MCs are a public health concern, especially during warmer months when the likelihood and frequency of human exposures through inhalation or ingestion of MC-contaminated waters increase [10,11,12]. Human MC exposures have been reported worldwide [13,14,15,16], with one of the most notable occurring at a kidney dialysis center in Brazil [17]. Patients were intravenously (IV) exposed to dialysate fluid contaminated with microcystin. Among the 101 exposed kidney dialysis patients who experienced liver failure, 50 died. 

One of the most well-studied, commonly detected, and toxic MC congeners is MC-LR [18,19]. On the basis of numerous human MC exposures worldwide, the World Health Organization has established a provisional limit of 1 µg/L of MC-LR equivalents in drinking water. In the United States, the Environmental Protection Agency has a 10-day health advisory limit of 1.6 µg/L MC-LR equivalents in drinking water for school-age children through to adults [20,21]. So far, at least 279 MC congeners have been identified. Most of those contain a conserved region called the Adda-side chain (3-amino-9-methoxy-2,6,8-trimethyl-10-phenyldeca-4,6-dienoic acid) [22,23]. This moiety is an essential component to MC toxicity, although the toxicity of the MC congeners varies substantially [5,24,25,26,27]. 

After ingestion, MCs are absorbed by the gastrointestinal tract [13] and subsequently taken up by liver cells [28,29]. Inside liver cells, MCs bind to and inhibit the enzymatic activity of protein phosphatase 1 and 2A (PP1 and PP2A) [30,31,32,33]. These phosphatases are essential for fundamental biological processes; inhibition results in defects in cytoskeletal integrity, leading to cellular toxicity [34]. The Adda moiety binds to a pocket within this catalytic site, contributing to PP1 and PP2A inhibition [32,33]. Assays measuring protein phosphatase inhibition, such as protein phosphatase inhibition assays (PPIA), can be used to determine variation in MC toxicity and screen samples for the presence of MCs in clinical specimens.

Previously described methods to detect MCs in water, urine, plasma, and serum samples include PPIA [35,36], enzyme-linked immunosorbent assays (ELISA) [37,38,39], gas chromatography–mass spectrometry (GC/MS) [37,39], and liquid chromatography coupled with tandem mass spectrometry (LC-MS/MS) [39,40,41,42]. These methods have varying levels of sensitivity and specificity, depending on the matrix and the use of targeted approaches such as LC-MS/MS or broader screening approaches such as PPIA [43]. 

The large variety of MCs poses challenges for monitoring specific congeners due to the lack of commercially available standards. To address this challenge, we extracted MC congeners from a sample using an antibody specific to the MC Adda side chain. We then analyzed the extract, using a commercially available PPIA kit to screen for potential exposures of MCs in human urine. The kit produces a response for any MCs that are recognized by the Adda antibody and inhibit PP2A. We also found that this method can detect nodularin. This ultimately provides more comprehensive data on potential exposures in which commercial standards are unavailable [35,40]. 

Here, we adapted this approach to detect MCs in human plasma. While most of the assay conditions remained the same, we optimized the MC capture time for this matrix and added a wash step. The wash step used phosphate-buffered saline (PBS) containing Tween 20 (PBS-T) to remove non-specifically bound proteins or particulates commonly found in human plasma. Without the combination of immunocapture and PBS-T wash steps, the plasma matrix effects caused a high background response with the PPIA kit, drastically limiting the method sensitivity. We also assessed the recovery of MC congeners from plasma and the capability of this method to detect unconjugated and conjugated forms of MCs, MC-LR-cysteine (MC-LR-Cys), and MC-LR-glutathione (MC-LR-GSH). We automated this assay using a robotic liquid handler, which enhances the capacity for high-volume sample analysis and minimizes user-dependent variabilities. Finally, we evaluated human plasma samples from self-reported exposures of direct or indirect contact with cyanobacterial blooms in Florida during 2018–2020. 

## 2. Results

### 2.1. Method Optimization

To evaluate the time required for MC-LR capture onto the Adda-antibody-coupled magnetic beads, we incubated 500 µL of plasma containing 1 ng/mL of MC-LR for 1, 5, 20, 30, 45, and 60 min (Figure 1). Following extraction from magnetic beads, the eluate was analyzed by LC-MS/MS. Average recovery was calculated after normalizing to a postfortification standard based on areas captured from LC-MS/MS (Figure 1). Analysis from Tukey’s multiple comparisons test showed no statistically significant differences between the 20 min timepoint (indicated in red, Figure 1) and the 30, 45, or 60 min incubation times. Statistical differences were seen between the 1, 5, and 20 min timepoints. Based on these results, a 20 min capture time was chosen and used for the manual and the automated steps in this method.

### 2.2. MC Immunocapture Recovery

Following immunocapture and PPIA, percent recoveries were calculated after normalization to postfortification solutions of elution buffer containing MC-LF, MC-LR, and MC-RR (Figure 2). Average recovery of the 0.050 ng/mL fortified concentration was 57.4%, 77.8%, and 51.2% for MC-RR, MC-LR, and MC-LF, respectively. The average recoveries of the 0.20 ng/mL fortified concentration were 63.5%, 57.6%, and 59.5% for MC-RR, MC-LR, and MC-LF, respectively. Our previous analysis showed that the elution buffer formulation containing 0.5% formic acid is compatible with these variations of charged congeners without interfering with the downstream PPIA [35]. 

### 2.3. Validation of MC-LR Detection Using PPIA in Human Plasma

To validate this method, 20 curves, with no more than two curves run per day, including anchors, calibrators, and quality control (QC) samples, were run manually or on the automated instrument using four different PP2A kit lots over the course of 29 weeks. The calibrator levels were prepared at 0.50, 0.30, 0.10, 0.050, and 0.030 ng/mL. Anchors were prepared at 1.0 and 0.010 ng/mL. QCs were prepared at 0.20 and 0.050 ng/mL (high and low, respectively). Based on these parameters, the method quantitative range was from 0.030–0.50 ng/mL. Interday (*n* = 20) percent relative standard deviation (RSD) for QCs varied between 7.18–15.8%. Intraday (*n* = 3) QC percent RSD varied between 8.70–18.6% (Table 1). We also evaluated 50 blank plasma samples to test for matrix interferences; all results were below the lowest calibrator (<0.03 ng/mL MC-LR equivalents) (data not shown). 

Method accuracy was evaluated using two different plasma pools fortified with three MC-LR concentrations (0.040, 0.080, and 0.20 ng/mL) and run in triplicate. Concentrations of the fortified samples were calculated using calibration standards that underwent the same procedure described in the prior immunocapture recovery section. Average percent accuracies for the 0.040, 0.080, and 0.20 ng/mL fortified concentrations were 99.6%, 82.5% and 101.8% for pool 1 and 90.0%, 86.9%, and 94.5% for pool 2, respectively (Figure 3). These data are within the current guidelines for ligand-binding assays, as outlined by the U.S. Food and Drug Administration [44].

### 2.4. Detection of Conjugated MCs

MC conjugates—MC-LR-Cys and MC-LR-GSH—were synthesized in-house and fortified into pooled human plasma. This method produced positive responses to MC-LR-Cys and MC-LR-GSH conjugates with average immunocapture values (black bars) of 0.839 and 0.436 ng/mL MC-LR equivalents, respectively. These data are not significantly different from post-spike concentrations (gray bars) (Figure 4).

### 2.5. Analysis of Plasma Samples from Florida Residents

As previously described, persons affected by a cyanobacterial algal bloom in South Florida provided urine, nasal swabs, and blood samples [35]. The collection of these samples followed approved protocols by the Florida Atlantic University Institution Review Board (# 929397-1). Here, we tested 188 plasma samples from 188 persons and found no positives. 

## 3. Discussion

To our knowledge, this is the first published report of an analytically validated method developed to detect MC in human plasma. Prior studies have detected MCs in mouse plasma, mouse and human urine, and mouse and human serum [35,37,39,41,45]. Animal models show that MC largely accumulates in the liver, with perfusion studies showing about 9% MC excretion in urine over 6 days after initial dosing [46]. For blood, in vitro mouse studies have also shown relatively rapid elimination of MCs in plasma, with half-lives at around 1–7 min [46]. In contrast with mouse studies, human studies detected unconjugated MCs in human serum for more than 50 days after acute exposure [37]. 

Several studies previously measured levels of MC exposure in mice. Palagma et al. (2018) orally administered two concentrations of MC-LR to mice over the course of 5 weeks and measured MC-LR levels in urine and plasma using LC-MS/MS. In plasma, this study found an average concentration of 0.33 ng/mL MC-LR for mice administered with 50 µg MC-LR/kg and an average concentration of 0.52 ng/mL MC-LR for mice administered with 100 µg MC-LR/kg. In urine, this study found an average concentration of 1.3 ng/mL MC-LR for the 50 µg MC-LR/kg group and an average concentration of 2.82 ng/mL MC-LR for the 100 µg MC-LR/kg group. Comparatively, Wharton et al. (2018) measured MC-LR concentrations using LC-MS/MS in mouse urine after intravenous injection of 40 µg MC-LR/kg and found an average of 22.8 ng/mL MC-LR. The order of magnitude difference in MC-LR urine concentrations between the studies is potentially due to different methods of MC administration (oral vs. intravenous) and doses (multiple doses vs. single injection) of MC-LR administration [40,41]. 

Mouse studies provide a controlled environment to assess MC exposures. However, to understand potential human exposures to MCs, we need clinical methods assessing human matrices. Hilborn et al. (2007) performed a serological evaluation from kidney dialysis patients intravenously injected with MCs from two separate instances in Brazil in 1996 (Caruaru) and 2001 (Rio de Janeiro). That study used ELISA to measure unconjugated MCs and GC/MS to measure total MCs. Total MCs were measured by determining the concentration of *erythro-2-methyl-3-methoxy-4-phenylbutyric* acid (MMPB). This chemical is derived from oxidizing unconjugated and conjugated MCs. Unconjugated MCs from Caruaru patients had a median serum concentration of 13.1 ng/mL; Rio de Janeiro patients had a median concentration of 0.34 ng/mL. Total serum MC concentration measurements were only performed on six Caruaru samples because of sample volume constraints, yielding a median of 52.8 ng/mL (compared with 13.1 ng/mL unconjugated MC). This might indicate that unconjugated MCs are a small subset of the total MCs present in serum [37]. 

Because acute exposures are uncommon, Wharton et al. (2019) developed a PPIA method to detect potential low-level MC inhalation from suspected exposures in human urine. To quantitate MCs using this method, we created a calibration curve with urine fortified with MC-LR, one of the most well-studied, commonly detected, and toxic MC congeners. This method has a lower reportable limit of 0.05 ng/mL MC-LR equivalents and detected three positives with an average of 0.065 ng/mL MC-LR equivalents [35]. These results are far below the concentrations detected from the exposed kidney dialysis patients. 

Ultimately, the concentrations determined from these studies show the need for a sensitive plasma method to compare the residence time of low-level MC inhalation exposures in urine and plasma. Our method achieves this by having a lower reportable limit of 0.03 ng/mL MC-LR equivalents and a limit of detection of 0.0183 ng/mL MC-LR equivalents. The limit of detection was calculated by the Clinical and Laboratory Standards Institute EP17 guidelines using the standard deviation of 50 plasma samples without known exposure from a convenience set and low-level spiked plasma samples to calculate the limit of detection [47,48]. The blank samples ranged from 0.00 to 0.013 with an average of 0.0011 and a standard deviation of 0.0034 ng/mL MC-LR equivalents. Other published methods have varying sensitivities (Table 2). 

After a human is exposed to MCs through consumption or inhalation, MCs interact with blood proteins, such as albumin, and might form conjugates with glutathione or other cysteine-containing compounds [46,49]. After exposure, it is unknown whether the unconjugated or conjugated form circulates at higher concentrations in blood [50]. When exposed blood is analyzed, targeted methods might not detect conjugated MCs, potentially resulting in underreported levels or false negative results. Furthermore, it has been shown that human serum albumin (HSA) demonstrates high affinity for MCs compared to other mammals and fish, indicating that humans may be more susceptible to toxic effects from MCs than other animals [51]. This could influence the predominant forms of MCs in circulation after exposure. To determine whether this method could detect conjugated MCs, we synthesized two MC conjugates, MC-LR-Cys and MC-LR-GSH, and fortified these materials in pooled plasma. The two synthesized MC conjugates produced positive responses, indicating that this method can detect unconjugated and conjugated forms of MCs. However, the MC-LR-Cys and -GSH produced reduced responses of 38 and 23% of the expected spiked concentrations of 2.2 and 1.9 ng/mL, respectively (Figure 4). Since these values are MC-LR equivalents, this indicates that the conjugates do not inhibit PP2A as strongly as unconjugated MC-LR. On the other hand, there were no significant differences between immunocapture and post-spike MC-LR conjugate detection; this indicates that the antibody cross reacts well with the MC-LR conjugates (Figure 4). These results could also indicate that this method may have a reduced ability to detect HSA-bound MCs in plasma. As discussed previously, HSA has a high affinity for MCs, and the variation in the forms of conjugated MCs would play a major role in human toxicity [51]. Although the measurement of HSA-bound MCs is not specifically evaluated here, this is an area for further study. Our analyses from algal blooms in Florida from 2018–2020 allow for a side-by-side comparison of urine and plasma samples collected from 188 persons who self-reported known, unknown, or potential MC exposure. MCs were not detected in any of the 188 plasma samples, including the three matched samples that were positive for MCs in urine from our previous study [35]. Because we also showed that this method detects conjugated MCs, we presume that any conjugated MCs in these plasma samples also were below levels of detection. Reasons for not finding positive results might include route of exposure and the rapid elimination of MCs from plasma. Moreover, one study showed that mice were relatively inefficient at metabolizing MC-LR to MC-LR-Cys [52], which could further indicate that at least for MC-LR-Cys, the levels might be extremely low in low-level exposure samples. 

So far, several hundred MC congeners have been identified [22,53]. While the use of LC-HRMS can provide an almost-comprehensive approach for identifying toxins in environmental samples, the equipment cost is prohibitive, requires specialized training, and might not be sensitive enough to confirm exposures for clinical samples. Our method combines immunocapture with an anti-Adda antibody and a PP2A assay, so that any toxic MC that contains the Adda moiety can be detected with high sensitivity. Comparatively, our method requires less specialized equipment, minimal training, and provides sample toxicity measurement in lieu of congener identification. 

A potential limitation of this method is that the antibodies targeting the Adda group might only recognize 80% of known congeners [54]. Other congeners, containing a variant of an Adda group, might also be toxic. Another limitation is that this method is a screening tool and needs to be combined with a confirmatory method, such as LC-MS/MS. Future work can study the effects of other potential interfering substances found in blood (i.e., substances released by cellular organs or medications used to treat the patient) on the performance of the method. Despite these questions, future investigations regarding MC detection will greatly benefit from use of our method and its automated approach for broad Adda-moiety containing MC detection.

## 4. Conclusions

We have developed a validated method to detect MCs in human plasma. Previous methods have focused on acute MC toxicity [37,39,40,41,45], whereas this method was developed to detect low-level MC exposures (e.g., inhalation). This method has greater sensitivity than our urine method [35], but we could not detect MCs in human plasma samples. These results support previous studies in which MC levels were higher in urine than in plasma, ultimately indicating that urine might be a better matrix for detecting potential low-level MC exposures (see, for example, Wharton, et al. [39,40] and Chen, et al. [45]). Moreover, we have shown that this method can be used to detect conjugated MCs. This ability further supports the utility of this method for analyzing clinical matrices for potential exposures. Finally, we have demonstrated the ability to automate this method. Automation increases sample preparation consistency between runs, decreases laboratorian exposure to potentially harmful substances, and reduces laboratory time needed to prepare samples. Overall, as freshwater harmful algal blooms and human exposures to these blooms become more common, this method may serve as a simple and robust assay to detect MC exposures in human plasma.

## 5. Materials and Methods

### 5.1. Chemicals and Materials

MC-LR and MC-RR certified reference standards and MC-LF standard (stored at −20 °C), the anti-Adda antibody (AD4G2, stored at 4 °C), and the kit for MCs/nodularins PP2A (part number 520032, stored at 4 °C) were purchased from Eurofins Abraxis (Warminster, PA, USA). Ammonium bicarbonate, PBS (pH 7.4), and PBS containing Tween 20 (PBS-T) were purchased from Sigma-Aldrich (St. Louis, MO, USA). Ultrapure 18.2 MΩ-cm reagent-grade filtered water was produced in-house using an Aqua Solutions purification system (Jasper, GA, USA). Protein LoBind 2.0 mL microcentrifuge tubes, 96-well Protein LoBind 1.0 mL plates, 96-well TwinTec Lo-Bind PCR plates, and a Thermomixer C were purchased from Eppendorf (Hauppage, NY, USA). Acetonitrile was purchased from Pharmaco (Dawsonville, GA, USA). Methanol, formic acid, and adhesive PCR sealing foil sheets were obtained from Fisher Scientific (Waltham, MA, USA). Dynabeads MyOne Streptavidin T1, Zebaspin > 7 K molecular weight cut-off 0.5 mL desalting columns, and EZ-link NHS-PEG4-biotin were purchased from Life Technologies (Grand Island, NY, USA). A 96-well plate magnet was purchased from V&P Scientific (San Diego, CA, USA). A microcentrifuge tube magnet and a Dynabeads Sample Mixer were purchased from Invitrogen (Waltham, MA, USA).

### 5.2. Human Plasma

Pooled and individual human plasmas were purchased from Tennessee Blood Services (Memphis, TN). These samples were from random persons. All personal identifiers had been removed and use of the samples was not deemed to constitute human subject research as defined by Health and Human Services 45-CFR 46.102 (e).

Plasma samples were collected from the same persons involved in our previous study [35]. Samples were collected and stored by Florida Atlanta University, as previously described, following approved Florida Atlantic University Institution Review Board protocols (#929397-1) [35]. 

### 5.3. Preparation of Calibrators and Quality Control Samples

Pooled human plasma was used to create calibrators and quality control (QC) samples. The MC-LR certified reference standard (14.6 µg/mL) was diluted to create a 50.0 ng/mL working stock in pooled plasma. This working stock was further diluted into 25 mL volumetric flasks to create calibrators at 0.50, 0.30, 0.10, 0.040, and 0.030; anchors at 1.0 and 0.010 ng/mL; and QC samples at 0.20 (QC high) and 0.050 (QC low) ng/mL. All calibrators, anchors, and QCs were aliquoted and stored at −80 °C in Protein LoBind tubes to avoid MC-LR adhesion to polypropylene [55]. We used 50 individual convenience sample blanks plasmas to evaluate potential interferences from this human matrix. 

For congener recovery experiments, MC-LR, MC-RR, and MC-LF standards (14.6, 12.7, and 100 µg/mL, respectively) were first diluted to 100 ng/mL in methanol. Then, working stock solutions were diluted again to a concentration of 5 ng/mL in methanol and fortified into pooled human plasma used for recovery experiments. Samples for immunocapture and extraction were made at 0.200 and 0.050 ng/mL in pooled human plasma. Percent recovery of each congener was calculated by normalizing the extracted samples to blank extracted plasma fortified at 2.00 and 0.500 ng/mL. The postfortification concentration was used because this method used 500 µL sample eluted into a final volume of 50 µL, creating a 10-fold concentration. Eluate was transferred to a 96-well TwinTec LoBind PCR plate, sealed with foil, and stored at −20 °C until ready for analysis by liquid chromatography coupled with tandem mass spectrometry (LC-MS/MS). These analyses were performed in triplicate. 

MC-LR was isolated on an Agilent 1290 liquid chromatograph (Santa Clara, CA) using an Acquity UPLC BEH C18 column (130 Å, 1.7 μm, 2.1 × 50 mm^2^, Waters, Milford, MA) maintained at 40 °C. The mobile phase (A) comprised 0.5% (*v*/*v*) formic acid in deionized water, and (B) comprised 0.5% (*v*/*v*) formic acid in acetonitrile. The elution pump program had a flow rate of 400 μL/min, starting with 15% B and increasing to 75% B from 0.00 to 4.50 min; the mobile phase was adjusted to 95% B from 4.50 to 4.60 min, held at 95% B from 4.60 to 5.00 min, then decreased to 15% B from 5.00 to 5.50 min, and held at 15% B from 5.50 to 6.50 min for a total run time of 6.50 min.

MC-LR was detected using an Agilent 6495b triple quadrapole mass spectrometer (Santa Clara, CA) with a Jet Stream electrospray ion (ESI) source operating in positive ion mode. The following parameters were used for all transitions: gas temperature, 170 °C; gas flow, 20 L/min; nebulizer, 50 psi; sheath gas temperature, 400 °C; sheath gas flow, 12 L/min; capillary, 5500 V positive; nozzle voltage, 2000 V positive; iFunnel high pressure RF, 100 V positive; iFunnel low pressure RF, 110 V positive. MC-LR transitions were detected with a precursor ion of 498.3 *m*/*z* and product ions of 135.1 and 103.2 *m*/*z*. The fragmentor energy is fixed at 380 for both transitions. The 135.1 *m*/*z* collision energy was 14 and the 103.2 *m*/*z* collision energy was 70. Data were acquired and analyzed using MassHunter software (Santa Clara, CA, USA). 

### 5.4. MC Immunocapture and Protein Phosphatase Inhibition Assay

The immunocapture of MCs was prepared as previously described [35]. Briefly, anti-Adda antibody was biotinylated using the EZ-link NHS-PEG4-biotin kit. To couple the biotinylated antibody with streptavidin-linked magnetic beads, a ratio of 1.0 µg of antibody to 5 µL of magnetic beads was briefly vortexed and then incubated at 800 rotations per minute (rpm) on a Thermomixer C in 30 s intervals of shaking followed by 30 s of rest for a total of 5 min. After two washes with PBS-T, the antibody-coupled magnetic beads were resuspended in PBS-T at 25 µL/sample and dispensed into a Protein LoBind plate in 25 µL aliquots. To prepare the plasma samples, aliquots of anchors, calibrators, and QCs were centrifuged at 10,000× *g* for 5 min to pellet particulates. The samples were combined with the previously aliquoted antibody-coupled beads and incubated at 1850 rpm on a Thermomixer C in 30 s intervals of shaking followed by 2 min of rest for a total of 20 min. 

To separate the antibody-coupled beads from the plasma, the plate was first centrifuged at 2000 rpm for 2 min at room temperature, then placed on top of a 96-well plate magnet. Plasma supernatant was removed by pipetting, leaving the antibody-coupled bead pellet. The plate was removed from the magnet and the antibody-coupled beads were washed once with 500 µL of PBS-T to remove any additional particulates, centrifuged for 2000 rpm for 2 min at room temperature, and placed again on the magnet to separate the beads. PBS-T supernatant was removed by pipetting and the plate was removed from the magnet. To elute bound MCs from the magnetic beads, 42.5 µL (70% Ultrapure water, 30% acetonitrile, and 0.05% formic acid, *v*/*v*/*v*) was added to the beads,; the plate was briefly vortexed to break up the pellet and then incubated at 1400 rpm on a Thermomixer C for a total of 2 min. The beads were again separated on the magnet and elution buffer was transferred into the PP2A assay plate. To each sample, 7.5 µL of a 1 M ammonium bicarbonate buffer was added to bring the pH of each sample to 6.0–8.0 pH and a final volume of 50 µL.

Reagents from the PP2A kit were equilibrated to room temperature and the reactions were performed according to manufacturer’s instructions. Briefly, the phosphatase was diluted with 3 mL of phosphatase dilution buffer and rotated on an Invitrogen Dynabeads Sample Mixer for 60 min at 10–20 rpm at room temperature. After hydration, 70 µL of the phosphatase solution was added to each buffered sample eluate in the PP2A assay plate, and then 90 µL of the chromogenic substrate was added. The wells were covered with adhesive PCR sealing foil sheets and incubated on the Thermomixer C for 30 min at 0 rpm at 37 °C. Following the incubation, 70 µL of stop solution was added to each sample well, gently mixed, and read on a BioTek Powerwave HT microplate spectrophotometer at 405 nm, and analyzed using Gen5 software (Winooski, VT, USA). 

### 5.5. MC Capture Time Optimization

To evaluate the optimal time required for MC-LR capture, 500 µL of 1 ng/mL MC-LR in plasma was incubated for various timepoints with anti-Adda antibody coupled magnetic beads. Each timepoint was analyzed in triplicate, and plasma samples were incubated at 1850 rpm for 30 s intervals in a Protein LoBind plate at 25 °C. Following immunocapture, the plate was centrifuged at 2000 rpm for 2 min, placed on a magnet to separate beads, and plasma was removed. Beads were washed with 500 µL of PBS-T and resuspended to remove any particulates, and the plate was centrifuged again at 2000 rpm for 2 min. The plate was placed on a magnet to separate beads and to remove PBS-T. A total of 50 µL of elution buffer (containing 70/30/0.5% *v*/*v*/*v* water/acetonitrile/formic acid) was added to wells containing beads. The plate was shaken on a Thermomixer C at 1400 rpm for 30 min in 60 s intervals of shaking/rest at 25 °C. After beads were separated from the elution buffer using a magnet, the eluate was transferred to a 96-well TwinTec LoBind PCR plate. As a reference, postfortification solutions using 45 µL of Elution Buffer and 5 µL of a 100 ng/mL MC-LR working stock were prepared and aliquoted into the 96-well TwinTec LoBind PCR plate. Again, the postfortification reference concentration of 10 ng/mL was used because the anchors, calibrators, QC materials, and all unknown samples are concentrated from a 500 µL sample and eluted into a final volume of 50 µL before PPIA, which is a 10-fold concentration. The PCR plate was sealed with foil and stored at −20 °C until ready for analysis by liquid chromatography coupled with tandem mass spectrometry (LC-MS/MS) and run as above in the congener recovery experiments.

### 5.6. Preparation of Samples for Immunocapture Recovery

This assay uses immunocapture with an Adda-specific antibody. The standard curve was generated based on concentrations of MC-LR; relative recoveries of other Adda-containing MCs were also evaluated. MC standards were chosen as representative congeners exhibiting double charge (MC-RR), single charge (MC-LR), and no charge (MC-LF). Standard materials were used to create plasma solutions containing MC-RR, MC-LR, and MC-LF in plasma at 0.200 and 0.050 ng/mL

### 5.7. Conjugated MC Synthesis, Confirmation, and Specificity

MC conjugates—MC-LR-cysteine (MC-LR-Cys) and MC-LR-glutathione (MC-LR-GSH)—were synthesized following Kondo et al. [56]. Briefly, a solution of cysteine (1.2 mg) or glutathione (3.1 mg) in 0.72 mL of 5% potassium carbonate was added to a vial with 1 mg of MC-LR. The mixture was left in a closed vial for 4 h at room temperature. The reaction was neutralized to pH 7 with of solution of 0.2 M hydrochloric acid. The solution was loaded onto a Phenomenex Strata-X 33 µm 60 mg reverse-phase solid-phase extraction sorbent after conditioning the stationary phase with methanol and water. After loading, the sorbent was rinsed with water and then eluted with methanol. The methanol fraction was used without further purification.

To verify the presence of the synthesized conjugates, samples were analyzed using a high-resolution Agilent 6546 Q-TOF mass spectrometer. The instrument was operated in auto-MS/MS mode controlled with Agilent’s MassHunter data acquisition software, version B.09.00. Analytes were ionized in positive ESI with an Agilent Jet Stream source using the same chromatography described above. MS/MS precursors with abundances above 1000 counts within the *m*/*z* 100–1000 mass range were automatically selected and fragmented by collision-induced dissociation at 60, 80, and 100 eVs. The instrument was externally calibrated with Agilent low concentration ESI tuning mix, and analysis run internally calibrated with purine and HP-0921 (hexakis [1H, 1H, 3H-tetrafluoropropoxy]phosphazine), both found in Agilent’s ESI-TOF reference mass solution kit.

Synthesized conjugates’ mass spectrometry parameters were also optimized on an Agilent 6495b to determine relative conjugate concentration and remnant MC-LR concentration. MC-LR-Cys transitions were detected with a precursor ion of 558.8 *m*/*z* and product ions of 135.2 and 103 *m*/*z*. The fragmentor energy was fixed at 380 for both transitions and the 135.2 *m*/*z* collision energy was 30, while the 103.1 *m*/*z* was 78. MC-LR-GSH transitions were detected with a precursor ion of 652.5 *m*/*z* and product ions of 135.1 and 103 *m*/*z*. The fragmentor energy was fixed at 380 for both transitions. The 135.1 *m*/*z* collision energy was 30 and the 103.1 *m*/*z* collision energy was 80. Tuning the conjugate stock solutions resulted in very high total ion chromatogram response. This response indicated that the estimated concentrations of these stock solutions were around 300 µg/mL when compared to MC-LR-ISTD.

Because the MC conjugates were highly concentrated in each solution, each conjugate solution was diluted 1:5000 in 50/50/0.1% *v*/*v* water/acetonitrile/formic acid to an estimated concentration of 60 ng/mL for each conjugate. The diluted conjugate samples were run as unknowns alongside a MC-LR calibration curve ranging from 0.25 to 25 ng/mL. Post-run analysis detected 0.45 and 0.56 ng/mL of MC-LR remaining in MC-LR-Cys and MC-LR-GSH, respectively. To ensure no false positives, the previously diluted MC conjugate solutions were fortified into pooled human plasma to achieve 0.015 ng/mL MC-LR. This level of MC-LR is half of the lowest calibrator for this method, ensuring that this method produced no false positives from MC-LR. These dilutions resulted in estimated concentrations of 2.2 and 1.9 ng/mL for MC-LR-Cys and MC-LR-GSH, respectively.

### 5.8. Assay Automation

This method was automated using the Tecan Freedom EVO 200 workstation (Tecan US, Morrisville, NC, USA) and controlled by Freedom EVOware Standard software. The workstation was equipped with an eight-channel Air Liquid handling arm (Air LiHa) for retrieving disposable pipet tips (DiTis), aspirating, and dispensing solutions; a robotic manipulator arm (RoMa) for the movement of plates; a Te-Shake for orbital shaking; and a MIO hotel incubator for covered temperature control.

Before method automation, several steps were performed offline, as described above: (1) the anchors, calibrators, and QCs were centrifuged at 10,000× *g* for 5 min; (2) the biotinylated anti-Adda antibody was coupled to streptavidin-linked magnetic beads; (3) the phosphatase was hydrated; and (4) all PP2A kit reagents, immunocapture reagents, the magnetic plate, a clean 1 mL 96-well Protein LoBind plate, 1 mL 96-well plate containing 600 µL of PBS-T, and the PP2A assay plate were placed in appropriate locations on the Tecan worktable. A custom script was written using EVOware to run the automation protocol, which initiates at the bead–antibody dispense step, as described above in the manual extraction.

The Tecan was programmed to aliquot 25 µL of the antibody-coupled magnetic beads into the Protein LoBind sample plate, followed by 500 µL of anchors, calibrators, and QCs. Because of shake-speed limitations of the onboard orbital shaker, the bead–antibody and plasma incubation mixing step was performed by pipetting instead of shaking as described in the manual extraction. Immediately after the plasma samples were added to the beads, the samples were mixed once via pipetting and incubated for 5 min at room temperature, continuing for 20 min. The RoMa then transferred the sample plate onto the magnetic plate, where samples incubated for 10 min at room temperature to capture the magnetic beads to the bottom of the plate. The plasma supernatant was aspirated from the sample plate and transferred to waste. After supernatant removal, the sample plate was removed from the magnet, 500 µL of PBS-T was added to each sample well, mixed once by pipetting, and transferred back onto the magnetic plate for an additional 2 min to separate the beads. The PBS-T supernatant was aspirated from the sample plate and transferred to waste. The sample plate was removed from the magnet and 42.5 µL elution buffer (prepared as described above) was added to each sample well. The sample plate was then transferred to an orbital shaker, mixed at 1400 rpm for 2 min, and returned to the magnetic plate for an additional minute to capture the magnetic beads. The elution buffer was transferred from the sample plate to the PP2A assay plate, followed by the addition of 7.5 µL of 1 M ammonium phosphate buffer. A total of 70 µL of phosphatase solution was added to each sample well of the PP2A assay plate, followed by 90 µL of chromogenic substrate. The PP2A assay plate was transferred to the hotel incubator for 30 min at 37 °C. After incubation, the PP2A assay plate was retrieved from the hotel and 70 µL of stop solution was added to each sample well. The plate was gently mixed by tapping, then read on the BioTek Powerwave HT microplate spectrophotometer as described above.

### 5.9. Data Analysis and Software

Absorbance data were analyzed using Gen5 v. 2.04 (BioTek, Winooski, WT, USA). Anchors and calibrators were analyzed using a 4-parameter curve. Statistical analyses were performed on GraphPad Prism 8 (La Jolla, CA, USA).

## Figures and Tables

**Figure 1 toxins-14-00813-f001:**
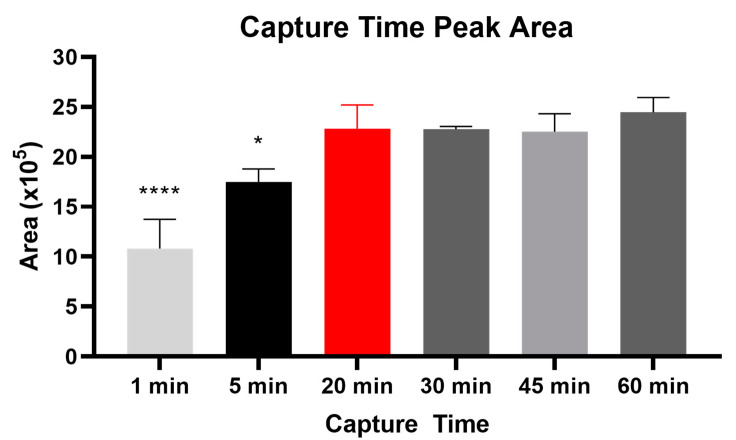
Comparison of antibody–analyte capture times by liquid chromatography coupled with tandem mass spectrometry area. Average area was calculated after normalizing to a postfortification standard. The red bar indicates the prescribed capture time. The asterisks indicate significance: **** represents a *p*-value < 0.001 and * represents a *p*-value < 0.05 when compared to the red bar (20 min capture time).

**Figure 2 toxins-14-00813-f002:**
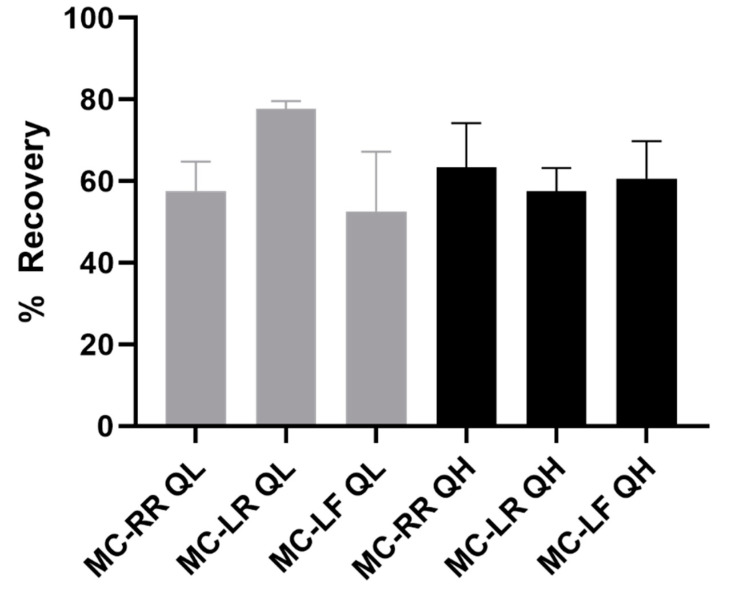
Percent recoveries of representative microcystins (MCs) after immunocapture. Recoveries were calculated after normalization to postfortification solutions (*n* = 3). Light gray bars represent recoveries at quality control low (QL, 0.0500 ng/mL) and black bars represent recoveries at quality control high (QH, 0.200 ng/mL).

**Figure 3 toxins-14-00813-f003:**
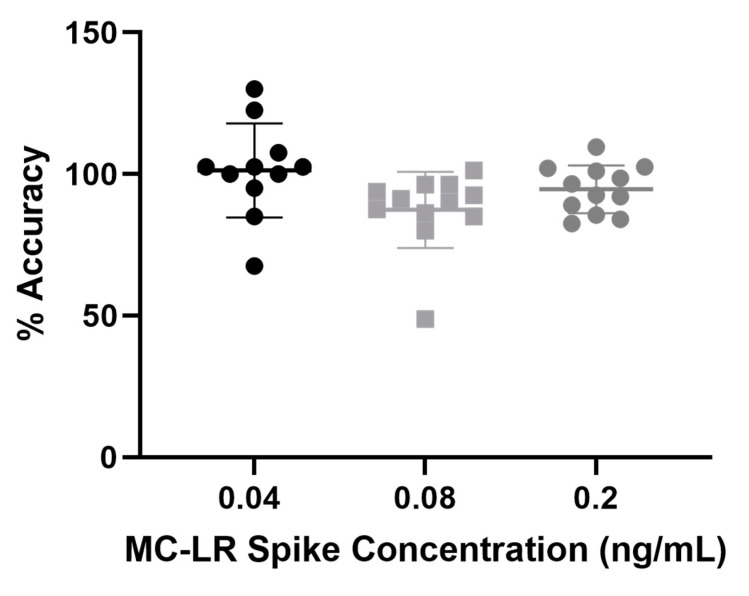
Method accuracy in two different plasma pools fortified with three microcystin-LR (MC-LR) concentrations (0.040, 0.080, and 0.20 ng/mL), run in triplicate, over 2 days.

**Figure 4 toxins-14-00813-f004:**
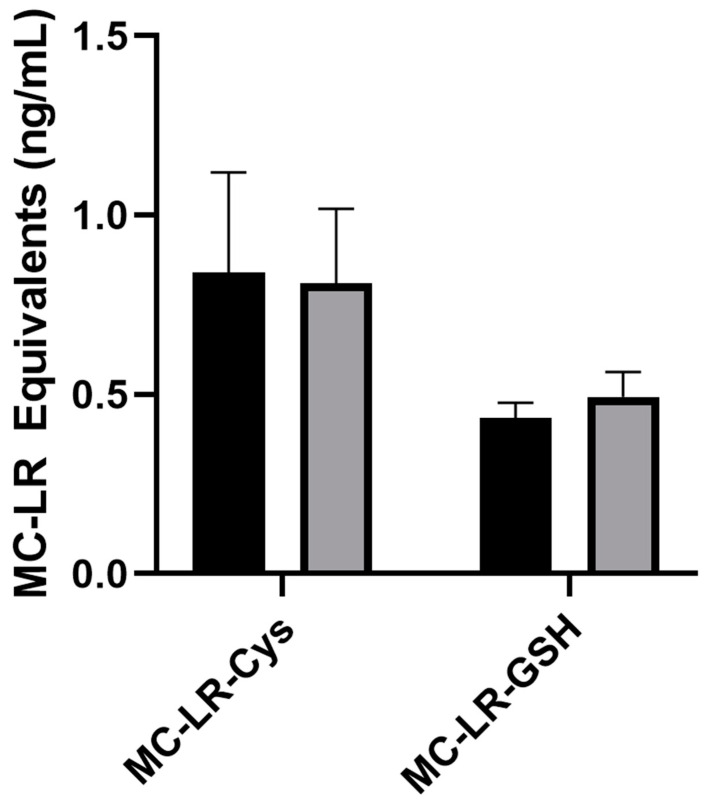
Method detection of conjugated microcystins (MCs). Pooled plasma fortified with MC-LR-cysteine (MC-LR-Cys) and MC-LR-glutathione (MC-LR-GSH) before immunocapture (black bars) were compared to 0.15 ng/mL post-spike controls (gray bars). Error bars represent the standard deviation of triplicate samples.

**Table 1 toxins-14-00813-t001:** Results from method validation. Interday (*n* = 20) percent accuracy and relative standard deviation (RSD) for calibrators and quality controls (QCs) and intraday (*n* = 3) percent accuracy and RSD for QCs.

	MC-LR (ng/mL)	Average (ng/mL)	Percent Accuracy	Percent RSD
Interday	0.50	0.521	104	10.7
	0.30	0.333	111	10.9
	0.20	0.228	114	7.18
	0.10	0.097	97	3.96
	0.05	0.0495	99	15.8
	0.04	0.0447	112	12.6
	0.03	0.032	107	22.0
Intraday	0.20	0.203	102	18.6
	0.05	0.0575	115	8.7

**Table 2 toxins-14-00813-t002:** Published method sensitivity comparison across human matrices.

Published Method	Analytically Validated?	Matrix	Limit of Detection	Lowest Reportable Limit	MC-LR Recovery
Measurement of Microcystin Activity in Human Plasma Using Immunocapture and Protein Phosphatase Inhibition Assay *	Yes	Human Plasma	0.0183 ng/mL	0.0300 ng/mL	77.8%
Measurement of microcystin and nodularin activity in human urine by immunocapture-protein phosphatase 2A assay [35]	Yes	Human Urine	0.0283 ng/mL	0.0500 ng/mL	77.6%
A simple colorimetric method to detect biological evidence of human exposure to microcystins [38]	No	Human Serum	0.147 ng/mL	0.500 ng/mL	N/A
Development and applications of solid-phase extraction and liquid chromatography-mass spectrometry methods for quantification of microcystins in urine, plasma, and serum [41]	No	Human Serum	Signal to noise of ~3	Signal to noise of ~10	98.4% (avg.)

* This manuscript.

## Data Availability

The data presented in this study are available in this article.

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
