# Peer review of "Measurement of Microcystin Activity in Human Plasma Using Immunocapture and Protein Phosphatase Inhibition Assay"

_toxins, 2022, doi:10.3390/toxins14110813_

Round 1

Reviewer 1 Report

I think it is a very good study, though it is losing some originality from the fact the immunocapture was already done in urine, and then is just applied again here to serum. It would have been more interesting and specific to the serum application, to see if the MC-LR-HSA conjugate, as another detoxification/free LR sink, could be captured and its PPIA effects in the context of the LR-GSH and LR-Cys conjugates. Perhaps it cannot be studied with this method, and leaves a major hole for future work regarding the release and analysis of LR from albumin.  In general, I understand there must be line drawn for which products to analyze. This is even acknowledged on line 216, and a similar study with BSA and the BSA-LR conjugate was studied on ELISA doi:10.1016/j.microc.2004.07.015 .  So the technology and precedent were both available, but not addressed.  This experiment should be done in the context of this article.

This LR-HSA experiment is important to the direct application of this method as well as clearing the path forward for the protein-conjugated MC sinks that account for congener specific half-lives.  We need to know if this works for at least the MC-LR-HSA before we can decide if we can use this method, or if we need to worry about further digestion/release mechanisms.

Author Response

Thank you for this comment. Determining the fate of MCs in humans is extremely important as well as quantifying MC-LR-HSA after potential exposure. However, this method is not intended to be a comprehensive/complete identification of exposure. Since MC protein plasma binding in mammals ranges from 16-28%, this would be a small fraction of the detectable MCs (Zhang et al, 2013). Also, if MC-LR was binding to albumin in plasma in large quantities, and this method could not detect MC-LR-HSA, our experimental recoveries would be low. Instead, we have MC-LR recoveries upwards of 79% (Figure 2).

Zhang, W., Liang, G., Wu, L. et al. Why mammals more susceptible to the hepatotoxic microcystins than fish: evidences from plasma and albumin protein binding through equilibrium dialysis. Ecotoxicology 22, 1012–1019 (2013). https://doi.org/10.1007/s10646-013-1086-5

Reviewer 2 Report

Journal: Toxins (ISSN 2072-6651)

Manuscript ID: toxins-1935682

Type: Article

Title: Measurement of Microcystin Activity in Human Plasma Using Immunocapture and Protein Phosphatase Inhibition Assay

Section: Marine and Freshwater Toxins

Special Issue: Cyanobacterial Toxin and Secondary Metabolite Detection, Fate and Toxicity Assessment

This study aimed to develop an immunocapture and protein phosphatase inhibition assay for measurement of microcystins (MCs) in human plasma. I have the following comments and suggestions for the authors to improve the quality of manuscript.

1. Abstract

Line 8

“At least 279 microcystins have been identified”

Introduction

Line 45

“So far, at least 286 MC congeners have been identified.”

Please unify it.

2. Lines 45-47

“Most of those contain a conserved region called the Adda-side chain (3-amino-9-methoxy-2,6,8-trimethyl-10-phenyldeca-4,6-dienoic acid) [20–22].”

Miles, C.; Stirling, D. Toxin Mass List COM V16.0 (Microcystin and Nodularin Lists and Mass Calculators for Mass Spectrometry of Microcystins, Nodularins, Saxitoxins and Anatoxins); 2019. Available online: http://dx.doi.org/10.13140/RG.2.2.12580.22402.

The reference 22 Miles and Stirling (2019) is not a formal publication. Please delete the citation.

3. Lines 58-63

“Previously described methods to detect MCs in water, urine, plasma, and serum samples include PPIA, enzyme-linked immunosorbent assays (ELISA), gas chromatography-mass spectrometry (GC/MS), and liquid chromatography coupled with tandem mass spectrometry (LC-MS/MS) [34–42].”

Please list the references for PPIA, ELISA, GC/MS, and LC-MS/MS separately.

4. Figure 1 and Table 1

The presentation of statistical analysis is complex and confusing. Please use different letters to show significant differences among groups (p < 0.05). For example, "a" and "b" or "bc" or "c" have significant differences, but "b" and "ab" or "bc" have no significant differences.

Please delete table 1.

5. Section “2.4. Detection of Conjugated MCs”

Lines 146-148

“MC conjugates, MC-LR-Cys and MC-LR-GSH, were synthesized in-house and fortified into pooled human plasma. This method produced positive responses to MC-LR-Cys and MC-LR-GSH conjugates (Figure 4).

The description is too simple. What are the actual concentrations of MC-LR-Cys and MC-LR-GSH? What are the detected concentrations? What do black and gray bars mean?

6. Section “2. Results”

Lines 159-160

“2.5. Analysis of Plasma Samples from Florida Residents”

“Here, we tested 188 plasma samples from 188 persons and found no positives.”

Please add analyses by ELISA and LC-MS.

7. Section “3. Discussion”

Lines 162-163

“To our knowledge, this is the first published report of an analytically validated method developed to detect MC in human plasma.”

This is certainly wrong. There are several methods for detecting MC in human plasma, including ELISA, LC-MS.

8. Lines 163-164

“We have also tested fortified serum samples and found the method fit for purpose (data not shown).”

Please add the data in the revised manuscript.

9. Lines 209-211

“Our method achieves this by having a lower reportable limit of 0.03 ng/mL MC-LR equivalents and a limit of detection of 0.0183 ng/mL MC-LR equivalents.”

Please present data of limit of detection of 0.0183 ng/mL MC-LR equivalents.

10. Please list a table and insert some discussion to compare the method you developed and previously published methods, including limit of detection, limit of quantitation, sensitivity and specificity.

11. In addition to MCs, cyanobacteria synthesize and release a wide range of other metabolites, including nodularins (NODs). Also, similar to MCs, NODs can also inhibit protein phosphatases. Can this method detect NODs?

Please read and cite the following papers.

Challenges of using blooms of Microcystis spp. in animal feeds: A comprehensive review of nutritional, toxicological and microbial health evaluation. https://doi.org/10.1016/j.scitotenv.2020.142319

Global geographical and historical overview of cyanotoxin distribution and cyanobacterial poisonings. https://doi.org/10.1007/s00204-019-02524-4

Author Response

  1. Abstract

Line 8

“At least 279 microcystins have been identified”

Introduction

Line 45

“So far, at least 286 MC congeners have been identified.”

Please unify it.

Changed to 279.

  1. Lines 45-47

“Most of those contain a conserved region called the Adda-side chain (3-amino-9-methoxy-2,6,8-trimethyl-10-phenyldeca-4,6-dienoic acid) [20–22].”

Miles, C.; Stirling, D. Toxin Mass List COM V16.0 (Microcystin and Nodularin Lists and Mass Calculators for Mass Spectrometry of Microcystins, Nodularins, Saxitoxins and Anatoxins); 2019. Available online: http://dx.doi.org/10.13140/RG.2.2.12580.22402.

The reference 22 Miles and Stirling (2019) is not a formal publication. Please delete the citation.

Removed.

  1. Lines 58-63

“Previously described methods to detect MCs in water, urine, plasma, and serum samples include PPIA, enzyme-linked immunosorbent assays (ELISA), gas chromatography-mass spectrometry (GC/MS), and liquid chromatography coupled with tandem mass spectrometry (LC-MS/MS) [34–42].”

Please list the references for PPIA, ELISA, GC/MS, and LC-MS/MS separately.

References are now listed separately.

  1. Figure 1 and Table 1

The presentation of statistical analysis is complex and confusing. Please use different letters to show significant differences among groups (p ï¼œ 0.05). For example, "a" and "b" or "bc" or "c" have significant differences, but "b" and "ab" or "bc" have no significant differences.

Please delete table 1.

For simplicity, we are focusing solely on the 20-minute timepoint and the statistical comparison showing why this time was selected. Additional text was added to the manuscript to clarify this figure. Table 1 removed.

  1. Section “2.4. Detection of Conjugated MCs”

Lines 146-148

“MC conjugates, MC-LR-Cys and MC-LR-GSH, were synthesized in-house and fortified into pooled human plasma. This method produced positive responses to MC-LR-Cys and MC-LR-GSH conjugates (Figure 4).

The description is too simple. What are the actual concentrations of MC-LR-Cys and MC-LR-GSH? What are the detected concentrations? What do black and gray bars mean?

The requested information is provided in Materials and Methods section 5.7. The caption of Figure 4 describes the black and gray bars, “Method detection of conjugated microcystins (MCs). Pooled plasma fortified with MC-LR-cysteine (MC-LR-Cys) and MC-LR-glutathione (MC-LR-GSH) to dilute remnant MC-LR to 0.015 ng/mL (half the lowest calibrator to prevent false positives) before immunocapture (black bars) were compared to 0.15 ng/mL post-spike controls (gray bars). Error bars represent the standard deviation of triplicate samples.”

  1. Section “2. Results”

Lines 159-160

“2.5. Analysis of Plasma Samples from Florida Residents”

“Here, we tested 188 plasma samples from 188 persons and found no positives.”

Please add analyses by ELISA and LC-MS.

We did not perform ELISA and LC-MS analyses. This samples were only tested by the PP2A method described in the manuscript. Developing and analytically validating ELISA and LC-MS methods for MC detection in human plasma would require the equivalent of two other manuscripts of analysis and method validation.

  1. Section “3. Discussion”

Lines 162-163

“To our knowledge, this is the first published report of an analytically validated method developed to detect MC in human plasma.”

This is certainly wrong. There are several methods for detecting MC in human plasma, including ELISA, LC-MS.

While other methods may have reported MC measurements in human plasma (a vast majority in human sera), these are not analytically validated and cannot be used for patient samples. We developed a clinically validated method to analyze patient samples in a regulated laboratory, meeting US requirements.

  1. Lines 163-164

“We have also tested fortified serum samples and found the method fit for purpose (data not shown).”

Please add the data in the revised manuscript.

Statement removed.

  1. Lines 209-211

“Our method achieves this by having a lower reportable limit of 0.03 ng/mL MC-LR equivalents and a limit of detection of 0.0183 ng/mL MC-LR equivalents.”

Please present data of limit of detection of 0.0183 ng/mL MC-LR equivalents.

As stated in the manuscript, the limit of detection was calculated by the Clinical and Laboratory Standards Institute EP17 guidelines using the standard deviation of 50 plasma samples without known exposure from a convenience set and low-level spiked plasma samples to calculate the limit of detection. Since 50 samples were used, the range, mean, and variability was included.

  1. Please list a table and insert some discussion to compare the method you developed and previously published methods, including limit of detection, limit of quantitation, sensitivity and specificity.

New table added and labeled as Table 2.

  1. In addition to MCs, cyanobacteria synthesize and release a wide range of other metabolites, including nodularins (NODs). Also, similar to MCs, NODs can also inhibit protein phosphatases. Can this method detect NODs?

Yes, the base of this method uses the immunocapture procedure and PP2A kit described previously (Wharton et al., 2019). This method can capture and detect NODs, which would translate to this method.

Please read and cite the following papers.

Challenges of using blooms of Microcystis spp. in animal feeds: A comprehensive review of nutritional, toxicological and microbial health evaluation. https://doi.org/10.1016/j.scitotenv.2020.142319

Global geographical and historical overview of cyanotoxin distribution and cyanobacterial poisonings. https://doi.org/10.1007/s00204-019-02524-4

References added.

Round 2

Reviewer 1 Report

You have now added citations for mouse serum but neither discuss the albumin conjugation, and are inferring the loss of MC-LR by loss of the UV/or 135 m/z peaks, which would not measure conjugated and unrecovered MC-LR. Additionally, mouse albumin detox site Cys34 is different than human Cys34 and this paper (specifically pointing toward MC-LR in Human serum) has to address the albumin conjugation if it is accounting for 15-20% of the conjugation and your recovery is 80%.  It is unreasonable to require a separate report on MC-LR-HSA, when you have presented a nearly a complete picture of MC-LR in human serum and the major detox products. All of the experimental procedures are already known, you simply have to complete Figure 4 to show that the PPIA does not measure MC-LR-HSA or the immunocapture cannot interact with ADDA enough when the MCLR is conjugated to HSA.  Alternatively, find citation which explains why that this is known for HSA or explain why this experiment was omitted.

Author Response

You have now added citations for mouse serum but neither discuss the albumin conjugation, and are inferring the loss of MC-LR by loss of the UV/or 135 m/z peaks, which would not measure conjugated and unrecovered MC-LR. Additionally, mouse albumin detox site Cys34 is different than human Cys34 and this paper (specifically pointing toward MC-LR in Human serum) has to address the albumin conjugation if it is accounting for 15-20% of the conjugation and your recovery is 80%.  It is unreasonable to require a separate report on MC-LR-HSA, when you have presented a nearly a complete picture of MC-LR in human serum and the major detox products. All of the experimental procedures are already known, you simply have to complete Figure 4 to show that the PPIA does not measure MC-LR-HSA or the immunocapture cannot interact with ADDA enough when the MCLR is conjugated to HSA.  Alternatively, find citation which explains why that this is known for HSA or explain why this experiment was omitted

The purpose of this method is to detect MCs in human plasma, which we have demonstrated in the manuscript. These data can stand alone without the need to determine MC-LR-HSA binding and immunocapture recoveries, which we believe is outside the scope of this manuscript.

Reviewer 2 Report

Journal: Toxins (ISSN 2072-6651)

Manuscript ID: toxins-1935682-peer-review-v2

Type: Article

Title: Measurement of Microcystin Activity in Human Plasma Using Immunocapture and Protein Phosphatase Inhibition Assay

Section: Marine and Freshwater Toxins

Special Issue: Cyanobacterial Toxin and Secondary Metabolite Detection, Fate and Toxicity Assessment

This study aimed to develop an immunocapture and protein phosphatase inhibition assay for measurement of microcystins (MCs) in human plasma. The quality of manuscript has improved during revisions. However, there are still some issues need to be addressed. I have the following comments and suggestions for the authors to improve the quality of manuscript.

1. Lines 89-95

“Analysis from Tukey’s multiple comparisons test showed no statistically significant differences between the 20-minute timepoint (indicated in red, Figure 1) and the 30-, 45-, or 60-minute incubation times. Statistical differences were seen between the 1-, 5-, 10-, and 20-minute timepoints. Additionally, there was no statistically significant difference between the 10- and 20- minute timepoints, but the 10- and 60- minute timepoints were significantly different.”

Figure 1

The presentation of statistical analysis is complex and confusing. Please use different letters to show significant differences among groups (p < 0.05). For example, "a" and "b" or "bc" or "c" have significant differences, but "b" and "ab" or "bc" have no significant differences.

I have made this comment last time. But the authors did not make any changes.

2. Figure 1

The data shows area. Please also present data of recovery.

3. Lines 107-110

“Average recovery of the 0.050 ng/mL fortified concentration was 57.4%, 77.8%, and 51.2% for MC-RR, MC-LR, and MC-LF, respectively. The average recoveries of the 0.20 ng/mL fortified concentration were 63.5%, 57.6%, and 59.5% for MC-RR, MC-LR, and MC-LF, respectively.”

Figure 2

The recovery is low. For quantitative analyses, recovery of 70%-130% is acceptable.

4. Figures 2 and 3, Table 1

Why the recovery is low in figure 2, 50%-80%, but the accuracy is about 100% and the recovery is about 100% in figure 3?

5. Lines 146-147

“This method produced positive responses to MC-LR-Cys and MC-LR-GSH conjugates (Figure 4).”

The description is too simple. What are the detected concentrations? What is the recovery? Please present the data.

I have made this comment last time. But the authors did not make any changes.

6. Table 2

Please add data of recovery.

Author Response

This study aimed to develop an immunocapture and protein phosphatase inhibition assay for measurement of microcystins (MCs) in human plasma. The quality of manuscript has improved during revisions. However, there are still some issues need to be addressed. I have the following comments and suggestions for the authors to improve the quality of manuscript.

  1. Lines 89-95

“Analysis from Tukey’s multiple comparisons test showed no statistically significant differences between the 20-minute timepoint (indicated in red, Figure 1) and the 30-, 45-, or 60-minute incubation times. Statistical differences were seen between the 1-, 5-, 10-, and 20-minute timepoints. Additionally, there was no statistically significant difference between the 10- and 20- minute timepoints, but the 10- and 60- minute timepoints were significantly different.”

Figure 1

The presentation of statistical analysis is complex and confusing. Please use different letters to show significant differences among groups (p ï¼œ 0.05). For example, "a" and "b" or "bc" or "c" have significant differences, but "b" and "ab" or "bc" have no significant differences.

I have made this comment last time. But the authors did not make any changes.

To reduce the complexity, we have reduced the unnecessary data points from Figure 1 and removed the phrase about 10 v 60 minutes in the text. Hopefully this provides clarity for the figure without adding more parts to the figure.

  1. Figure 1

The data shows area. Please also present data of recovery.

Figure 1 is intended to demonstrate the minimum time needed to capture the maximal amount of MC. Therefore, area is presented to demonstrate maximum recovery as a specific value.

  1. Lines 107-110

“Average recovery of the 0.050 ng/mL fortified concentration was 57.4%, 77.8%, and 51.2% for MC-RR, MC-LR, and MC-LF, respectively. The average recoveries of the 0.20 ng/mL fortified concentration were 63.5%, 57.6%, and 59.5% for MC-RR, MC-LR, and MC-LF, respectively.”

Figure 2

The recovery is low. For quantitative analyses, recovery of 70%-130% is acceptable.

Although the recovery is below the FDA recommended guidelines, the method is fit for purpose and deemed acceptable due to sensitive levels and lack of available clinical samples.

  1. Figures 2 and 3, Table 1

Why the recovery is low in figure 2, 50%-80%, but the accuracy is about 100% and the recovery is about 100% in figure 3?

The y-axis on Figure 3 is labeled incorrectly. The axis is re-labeled to “% Accuracy”.

  1. Lines 146-147

“This method produced positive responses to MC-LR-Cys and MC-LR-GSH conjugates (Figure 4).”

The description is too simple. What are the detected concentrations? What is the recovery? Please present the data.

I have made this comment last time. But the authors did not make any changes.

Additional information describing the data were added to the text and the Figure 4 caption was clarified. Recovery was not determined since we do not know specific starting concentration of the conjugates. Although we do not know specific concentration, we do know it is effective in capturing the conjugates.

  1. Table 2

Please add data of recovery

A recovery column was added.

Round 3

Reviewer 1 Report

I believe that without this HSA-MC-LR adduct added the paper is incomplete and we simply disagree on this point so there appears to be no resolution.  I do not understand why the GSH-MC-LR and Cys-MC-LR are in the scope, but the final large portion, the HSA-MC-LR which reportedly would account the remaining recovery, is not.

Author Response

We have added a Zhang et al. 2013 citation and further discussed albumin binding in the context of our method (lines 251-255 and 259-270).

Reviewer 2 Report

Journal: Toxins (ISSN 2072-6651)

Manuscript ID: toxins-1935682-peer-review-v3

Type: Article

Title: Measurement of Microcystin Activity in Human Plasma Using Immunocapture and Protein Phosphatase Inhibition Assay

Section: Marine and Freshwater Toxins

Special Issue: Cyanobacterial Toxin and Secondary Metabolite Detection, Fate and Toxicity Assessment

This study aimed to develop an immunocapture and protein phosphatase inhibition assay for measurement of microcystins (MCs) in human plasma. The quality of manuscript has improved during revisions. However, there are still some issues need to be addressed. I have the following comments and suggestions for the authors to improve the quality of manuscript.

1. Figure 1

The data shows area. Please also present data of recovery. This will help readers better understand the data.

I have made this comment last time. But the authors did not make any changes.

2. Lines 111-114

“Average recovery of the 0.050 ng/mL fortified concentration was 57.4%, 77.8%, and 51.2% for MC-RR, MC-LR, and MC-LF, respectively. The average recoveries of the 0.20 ng/mL fortified concentration were 63.5%, 57.6%, and 59.5% for MC-RR, MC-LR, and MC-LF, respectively.”

Figures 2 and 3, Table 1

Why the recovery is low in figure 2, 50%-80%, but the accuracy is about 100% in figure 3?

3. Lines 158-161

“This method produced positive responses to MC-LR-Cys and MC-LR-GSH conjugates with average immunocapture values (black bars) of 0.839 and 0.436 ng/mL MC-LR equivalents, respectively. These data are not significantly different from post-spike concentrations (gray bars) (Figure 4).”

What is the recovery? Please present the data. I have made this comment last time. But the authors did not make any changes.

What are the differences between immunocapture values and post-spike concentrations? Please add the information in the revised manuscript.

Author Response

  1. Figure 1

The data shows area. Please also present data of recovery. This will help readers better understand the data.

I have made this comment last time. But the authors did not make any changes.

We clarified this difference in the text.

  1. Lines 111-114

“Average recovery of the 0.050 ng/mL fortified concentration was 57.4%, 77.8%, and 51.2% for MC-RR, MC-LR, and MC-LF, respectively. The average recoveries of the 0.20 ng/mL fortified concentration were 63.5%, 57.6%, and 59.5% for MC-RR, MC-LR, and MC-LF, respectively.”

Figures 2 and 3, Table 1

Why the recovery is low in figure 2, 50%-80%, but the accuracy is about 100% in figure 3?

We have addressed this by clarifying the differences between “accuracy” and ”recovery” in the text.

  1. Lines 158-161

“This method produced positive responses to MC-LR-Cys and MC-LR-GSH conjugates with average immunocapture values (black bars) of 0.839 and 0.436 ng/mL MC-LR equivalents, respectively. These data are not significantly different from post-spike concentrations (gray bars) (Figure 4).”

What is the recovery? Please present the data. I have made this comment last time. But the authors did not make any changes.

What are the differences between immunocapture values and post-spike concentrations? Please add the information in the revised manuscript.

We have added additional information in the discussion to address this.